# Effect of cancer on outcome of COVID-19 patients: a systematic review and meta-analysis of studies of unvaccinated patients

Giulia Di Felice[1,2], Giovanni Visci[1,2]*, Federica Teglia[1], Marco Angelini[1], Paolo Boffetta[1,3]*

[1]Department of Medical and Surgical Sciences, University of Bologna, Bologna, Italy; [2]IRCCS Azienda Ospedaliero-Universitaria di Bologna, Bologna, Italy; [3]Stony Brook Cancer Center, Stony Brook University, Stony Brook, United States

*For correspondence:
giovanni.visci6@gmail.com (GV);
paolo.boffetta@unibo.it (PB)

**Competing interest:** The authors declare that no competing interests exist.

## Abstract

**Background:** Since the beginning of the SARS-CoV-2 pandemic, cancer patients affected by COVID-19 have been reported to experience poor prognosis; however, a detailed quantification of the effect of cancer on outcome of unvaccinated COVID-19 patients has not been performed.

**Methods:** To carry out a systematic review of the studies comparing the outcome of unvaccinated COVID-19 patients with and without cancer, a search string was devised which was used to identify relevant publications in PubMed up to December 31, 2020. We selected three outcomes: mortality, access to ICU, and COVID-19 severity or hospitalization. We considered results for all cancers combined as well as for specific cancers. We conducted random-effects meta-analyses of the results, overall and after stratification by region. We also performed sensitivity analyses according to quality score and assessed publication bias.

**Results:** For all cancer combined, the pooled odds ratio (OR) for mortality was 2.32 (95% confidence interval [CI] 1.82–2.94, $I^2$ for heterogeneity 90.1%, 24 studies), that for ICU admission was 2.39 (95% CI 1.90–3.02, $I^2$ 0.0%, 5 studies), that for disease severity or hospitalization was 2.08 (95% CI 1.60–2.72, $I^2$ 92.1%, 15 studies). The pooled mortality OR for hematologic neoplasms was 2.14 (95% CI 1.87–2.44, $I^2$ 20.8%, 8 studies). Data were insufficient to perform a meta-analysis for other cancers. In the mortality meta-analysis for all cancers, the pooled OR was higher for studies conducted in Asia than studies conducted in Europe or North America. There was no evidence of publication bias.

**Conclusions:** Our meta-analysis indicates a twofold increased risk of adverse outcomes (mortality, ICU admission, and severity of COVID-19) in unvaccinated COVID-19 patients with cancer compared to COVID-19 patients without cancer. These results should be compared with studies conducted in vaccinated patients; nonetheless, they argue for special effort to prevent SARS-CoV-2 infection in patients with cancer.

**Funding:** No external funding was obtained.

## Editor's evaluation

The authors conducted a systematic review and baseline meta-analysis of studies on the impact of SARS-Cov-2 infection on morbidity and mortality among cancer patients not previously vaccinated against the virus. This analysis serves as benchmark for forthcoming work on the same outcomes among vaccinated cancer patients, which as a whole will assist the development of cancer care guidelines.

## Introduction

Since the emergence of SARS-CoV-2, many studies have been conducted on the outcomes of COVID-19, in order to identify factors associated with a higher death rate and a more severe infection course. Some groups of patients at increased risk of severe COVID-19, morbidity, and mortality have been identified, including elderly patients, and those with comorbidities, such as hypertension, diabetes, chronic kidney disease, or COPD (*Fang et al., 2020*). Cancer patients are also a high-risk group due to their compromised immune systems and vulnerability to infection resulting from their disease and the treatments (*Kamboj and Sepkowitz, 2009*).

It is generally assumed that cancer patients are at higher risk for severe COVID-19 and death attributed to COVID-19 (*Rüthrich et al., 2021*). However, cancer encompasses a very heterogeneous group of diseases with a diverse range of subtypes and stages. In addition, not all cancers are equal in terms of incidence, prognosis, and treatment. This must be taken into account when the type of cancer is not specified (*Lee et al., 2020*). For this reason, although descriptions and analyses of risk factors, clinical courses, and mortality in cancer patients infected with SARS-CoV-2 have been reported, a quantitative assessment of the effect of cancer in patients with COVID-19 would be important to guide clinical decision-making.

We aimed at conducting a systematic review of the epidemiological features of the studies of COVID-19 in cancer patients conducted before the implementation of vaccination campaigns, and to provide a quantitative estimate of the risk of severe infection course and mortality in COVID-19 patients with cancer compared to COVID-19 patients without cancer. We decided to restrict our review to studies of unvaccinated patients because (i) they provide the clearest picture of the effect of cancer on outcome of COVID-19 patients, and (ii) the full effect of vaccination might not have been yet captured by available studies.

## Materials and methods

This systematic review was conducted according to the PRISMA statement (*Moher et al., 2009*). We submitted the protocol (available as *Supplementary file 1*) to the PROSPERO Registry. To carry out the systematic review of the scientific literature, the following string was used for the PubMed database:

(neoplas*[TIAB] OR tumor*[TIAB] OR cancer*[TIAB] OR malignancy [TIAB]) AND (2019 novel coronavirus[TIAB] OR COVID-19[TIAB] OR COVID19[TIAB] OR SARS-CoV-2[TIAB] OR 2019-nCoV[TIAB]).

In order to restrict the review to study populations on unvaccinated COVID-19 patients, we included papers published in peer-reviewed journals up to December 31, 2020. We excluded abstracts and non-peer-reviewed reports, articles in languages other than English, and studies including children. We also excluded reviews, meta-analysis and case reports, and studies with less than 50 patients or less than 10 events. Finally, we excluded studies in which diagnosis of SARS-CoV-2 infection was not made by PCR testing.

The articles were independently reviewed and abstracted by two pairs of reviewers (GDF and MA; GV and FT), on the basis of title, abstract, and full text; the disagreement between the authors of the reviews (6.1% of all studies) and was resolved through discussion with a fifth reviewer [PB].

We selected the following outcomes: mortality, ICU admission, severity of COVID-19 symptoms, and hospitalization: we combined these latter two outcomes because the definition of severity was heterogeneous across studies and the number of available studies was low. We excluded from the review studies addressing the impact of SARS-CoV-2 infection on prevention, diagnosis, and treatment of cancer patients, for example, studies comparing cancer patients with and without SARS-CoV-2 infection, as well as studies on the oncogenic effect of the virus, for example, analyses of cancer-related alterations. In addition, we carried out a back-search by inspecting the lists of references of articles selected for the review.

*Figure 1* shows the flowchart for selection of the studies. Details on the studies retained in each step of the process are available from the authors.

We abstracted the following parameters from the articles retained for the review: country, sample size, number of persons affected by cancer and by SARS-CoV-2 infection, cancer type and comparison group (patients without cancer or patients with a different type of cancer), outcome, and risk estimate (relative risk or odds ratio [OR]) with 95% confidence interval (CI). If the risk estimate or the

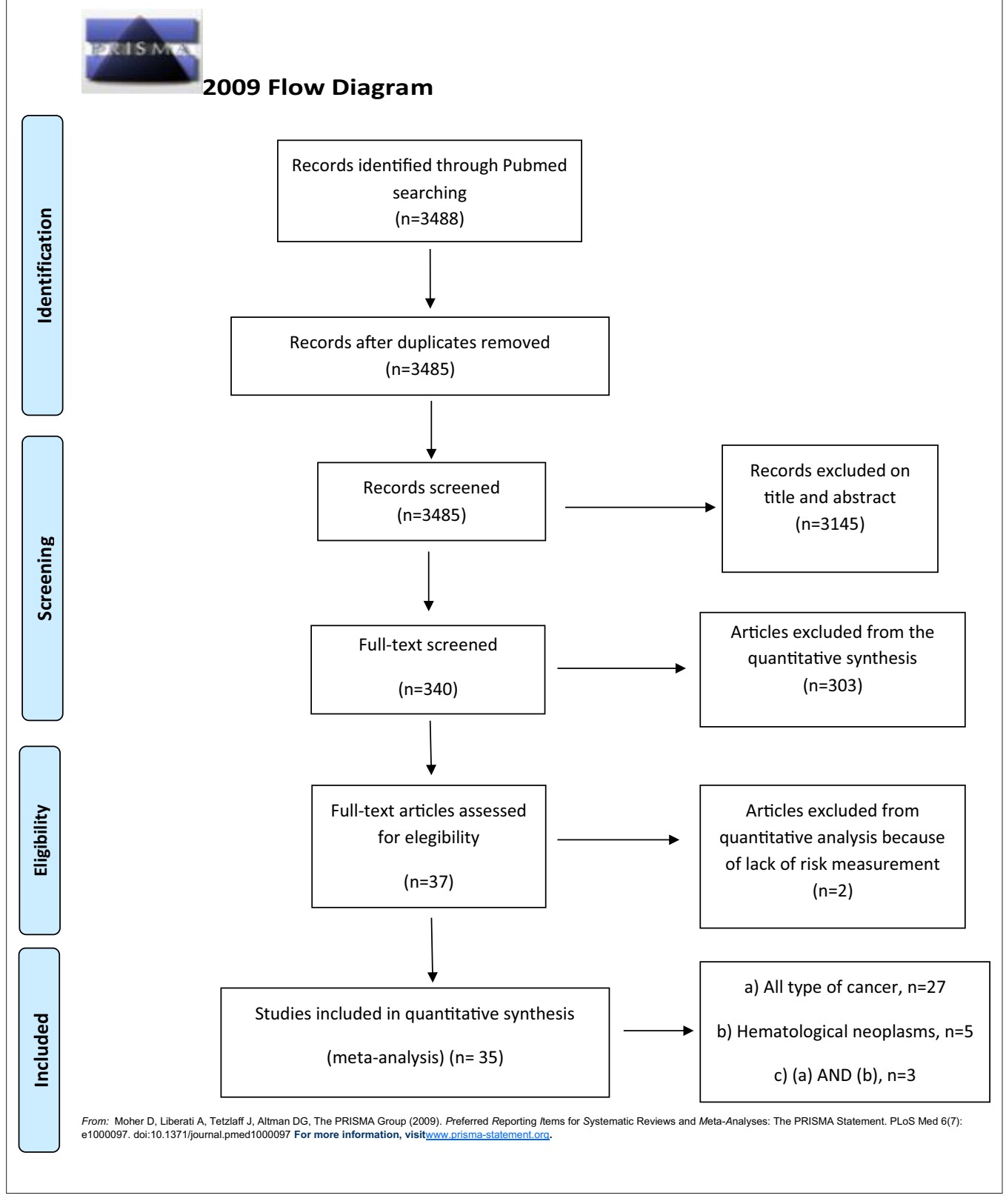

**Figure 1.** Flowchart for the identification of articles for the meta-analyses (PRISMA).

**Table 1.** Selected characteristics of studies included in the meta-analysis – all cancers.

| References | Country | N patients | N outcomes | Quality score | Comparison | Outcome | RR/OR* | 95% CI |
|---|---|---|---|---|---|---|---|---|
| **Dai et al., 2020** | China | 641 | 105 | 9.25 | Internal | Mortality | 2.34 | 1.15–4.77 |
| | | | | | | ICU | 2.84 | 1.59–5.08 |
| | | | | | | Severity | 2.79 | 1.75–4.44 |
| **Haase et al., 2021** | Denmark | 323 | 15 | 11.5 | Internal | Mortality | 3.18 | 1.66–6.09 |
| **Meng et al., 2020** | China | 436 | 109 | 9.5 | Internal | Mortality | 2.98† | 1.76–5.05 |
| **Sun et al., 2020** | USA | 323 | 67 | 10.5 | Internal | Mortality | 5.67† | 1.49–21.58 |
| | | | | | | ICU | 1.91† | 0.90–4.06 |
| | | | | | | Hospitalization | 2.16† | 1.12–4.17 |
| **Nogueira et al., 2020** | Portugal | 20,293 | 611 | 9.25 | Internal | Mortality | 1.48 | 1.07–2.05 |
| **Zandkarimi et al., 2020** | Iran | 1831 | 32 | 6.5 | Internal | Mortality | 3.57 | 1.82–7.02 |
| **Zhao et al., 2020** | China | 539 | 23 | 6.5 | Internal | Mortality | 3.23 | 1.39–7.51 |
| **Harrison et al., 2020** | USA | 31,461 | 1966 | 9.75 | Internal | Mortality | 0.87† | 0.72–1.09 |
| **Gupta et al., 2020** | USA | 2215 | 112 | 7.5 | Internal | Mortality | 2.20 | 1.50–3.22 |
| **Ganatra et al., 2020** | USA | 2476 | 195 | 9.25 | Internal | Mortality | 3.53 | 2.95–4.23 |
| | | | | | | Severity | 3.75 | 3.17–4.44 |
| | | | | | | Hospitalization | 2.78 | 2.37–3.26 |
| **Westblade et al., 2020** | USA | 2294 | 100 | 8.5 | Internal | Mortality | 1.29 | 1.04–1.61 |
| | | | | | | Severity | 0.76 | 0.57–1.01 |
| **Wang et al., 2020** | USA | 16,570 | 1200 | 11 | Internal | Mortality | 3.20 | 2.89–3.55 |
| | | | | | | Hospitalization | 2.85 | 2.63–3.09 |
| **Cherri et al., 2020** | Italy | 2039 | 53 | 11 | Internal | Mortality | 2.22† | 1.25–3.94 |
| **Görgülü and Duyan, 2020** | Turkey | 483 | 75 | 11 | Internal | Mortality | 1.81† | 0.88–3.72 |
| | | | | | | ICU | 2.14 | 1.26–3.63 |
| **Jiménez et al., 2020** | Spain | 1549 | 103 | 9.75 | Internal | Mortality | 4.29† | 2.40–7.67 |
| **Thompson et al., 2020** | UK | 470 | 87 | 10 | Internal | Mortality | 2.20† | 1.27–3.81 |
| **Li et al., 2020** | China | 1859 | 65 | 9 | External | Mortality | 1.59† | 0.94–2.68 |
| **Shoumariyeh et al., 2020** | Germany | 78 | 39 | 8.5 | External | Mortality | 1.01 | 0.41–2.49 |
| | | | | | | Severity | 1.15 | 0.61–2.17 |
| **Mehta et al., 2020** | USA | 1308 | 218 | 8.5 | External | Mortality | 2.38 | 1.69–3.35 |

*Table 1 continued on next page*

*Table 1 continued*

| References | Country | N patients | N outcomes | Quality score | Comparison | Outcome | RR/OR* | 95% CI |
|---|---|---|---|---|---|---|---|---|
| *Rogado et al., 2020* | Spain | 42,495 | 45 | 8 | External | Mortality | 4.82 | 2.67–8.71 |
| *Brar et al., 2020* | USA | 585 | 117 | 9.5 | External | Mortality | 0.98 | 0.58–1.66 |
| | | | | | | Severity | 0.80 | 0.57–1.13 |
| *Zhang et al., 2020b* | China | 217 | 112 | 9.5 | External | Mortality | 4.83 | 2.87–8.12 |
| | | | | | | ICU | 2.60 | 1.87–3.62 |
| | | | | | | Severity | 1.43 | 1.09–1.87 |
| *Sorouri et al., 2020* | Iran | 159 | 53 | 10 | External | Mortality | 3.27† | 0.93–11.55 |
| | | | | | | ICU | 1.52† | 0.56–4.12 |
| *Lunski et al., 2021* | USA | 5145 | 312 | 9.5 | External | Mortality | 2.03† | 1.44–2.87 |
| *Atalla et al., 2021* | USA | 339 | 27 | 10.5 | Internal | Hospitalization | 3.34 | 1.81–6.16 |
| *Cheng et al., 2020* | China | 1476 | 29 | 6 | Internal | Severity | 2.14 | 0.97–4.73 |
| *Song et al., 2021* | China | 961 | 21 | 7 | Internal | Severity | 2.77 | 1.16–6.62 |
| *Liang et al., 2020* | China | 1590 | 18 | 8.75 | Internal | Severity | 4.07† | 1.23–13.45 |
| *Bauer et al., 2021* | USA | 1449 | 108 | 8.5 | Internal | Severity | 1.72† | 1.11–2.67 |
| *Tian et al., 2020* | China | 751 | 232 | 9 | External | Severity | 3.75 | 2.71–5.19 |

*Results derived from data reported in the publication are in italics.

†Risk estimates derived from multivariate analysis.

RR: Relative Risk. OR: Odds Ratio. CI: Confidence interval. ICU: Intensive Care Unit.

**Table 2.** Selected characteristics of studies included in the meta-analysis – *hematological tumors*.

| Author – Year | Country | Sample size (n) | Cancer (n) Or number of cancer (%) | Quality assessment (CASP) | Comparison | Outcome | RR/OR* | IC |
|---|---|---|---|---|---|---|---|---|
| *Dai et al., 2020* | China | 641 | 9 | 9.25 | Internal | Mortality | 9.07 | 2.16–38.13 |
| *Haase et al., 2021* | Denmark | 323 | 13 | 11.5 | Internal | Mortality | 1.83 | 0.85–3.93 |
| *Meng et al., 2020* | China | 327 | 16 | 9.5 | Internal | Mortality | 2.83† | 0.96–8.32 |
| *Yigenoglu et al., 2021* | Turkey | 1480 | 740 | 10.5 | External | Mortality | 2.20 | 1.93–2.50 |
| *Shah et al., 2020* | UK | 1183 | 68 | 9.75 | External | Mortality | 1.74† | 1.12–2.71 |
| *Sanchez-Pina et al., 2020* | Spain | 92 | 39 | 10.25 | External | Mortality | 6.65† | 1.87–23.67 |
| *Passamonti et al., 2020* | Italy | 536 | 11 | 11 | External | Mortality | 2.04† | 1.77–2.35 |
| *Cattaneo et al., 2020* | Italy | 204 | 102 | 9 | External | Mortality | 2.10 | 1.14–3.85 |

*italics characters when calculated manually.

†Risk estimates derived from multivariate analysis.

CASP: Critical Appraisal Skills Programme. RR: Relative Risk. OR: Odds Ratio. IC: Interval Confidence. ICU: Intensive Care Unit.

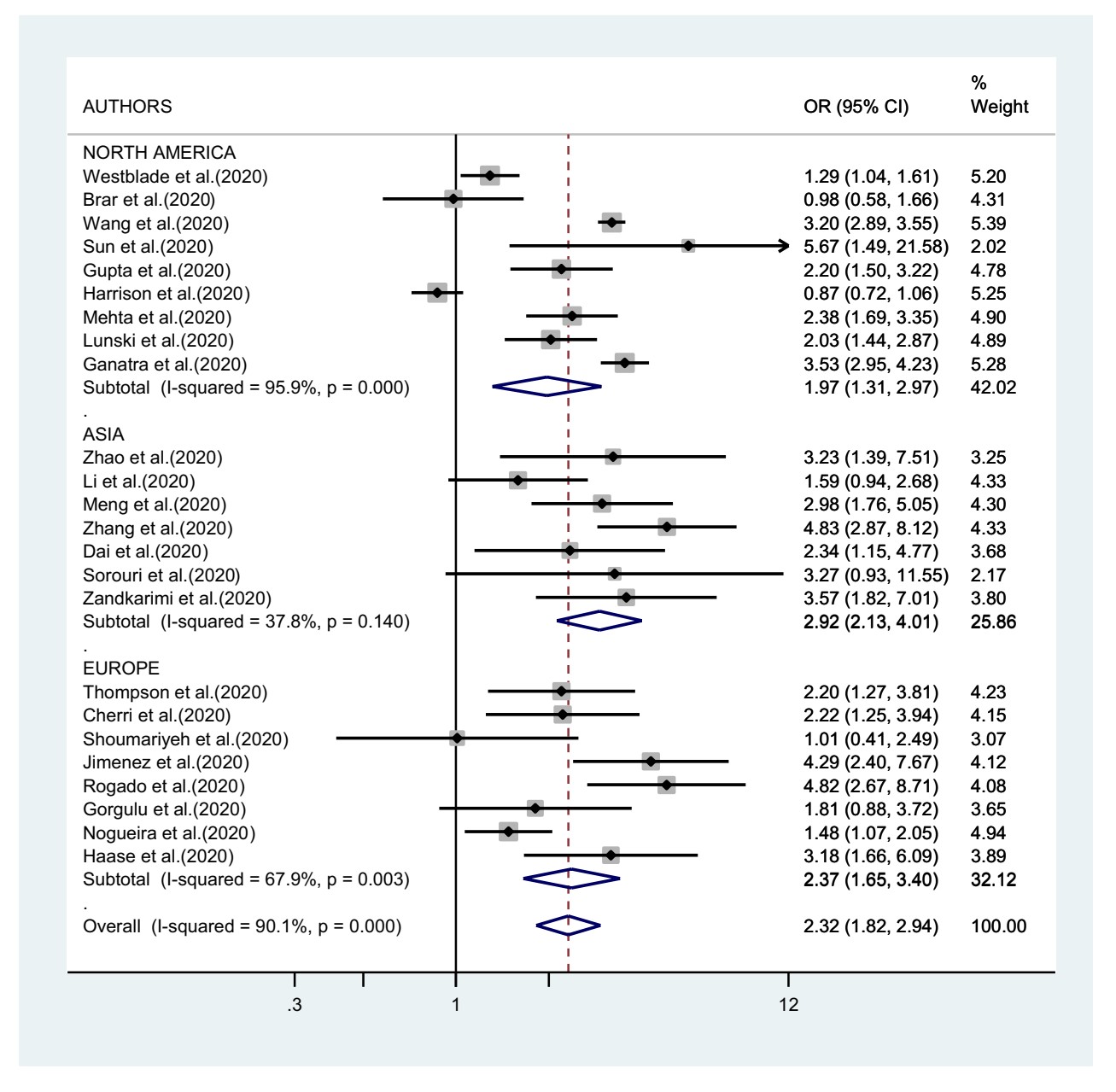

**Figure 2.** Forest plot – all types of cancer – outcome 1.

CI were not reported in the publication, we calculated them from the raw data, if possible. We also performed a quality assessment (QA) based on a modified version of CASP score (*Critical Appraisal Skills Programme, 2018*), that included 10 criteria (*Supplementary file 2*).

## Statistical analysis

We conducted random-effects (*DerSimonian and Laird, 1986*) meta-analyses of the risk estimates for the combinations of cancers and outcomes with more than five independent results. We also conducted stratified meta-analyses according to geographic region, to explore potential sources of heterogeneity, that we quantified using the $I^2$ test (*Higgins and Thompson, 2002*).

To evaluated results stability, we performed sensitivity analyses by quality score and repeated the meta-analysis after excluding one study at a time. We also conducted secondary analyses excluding

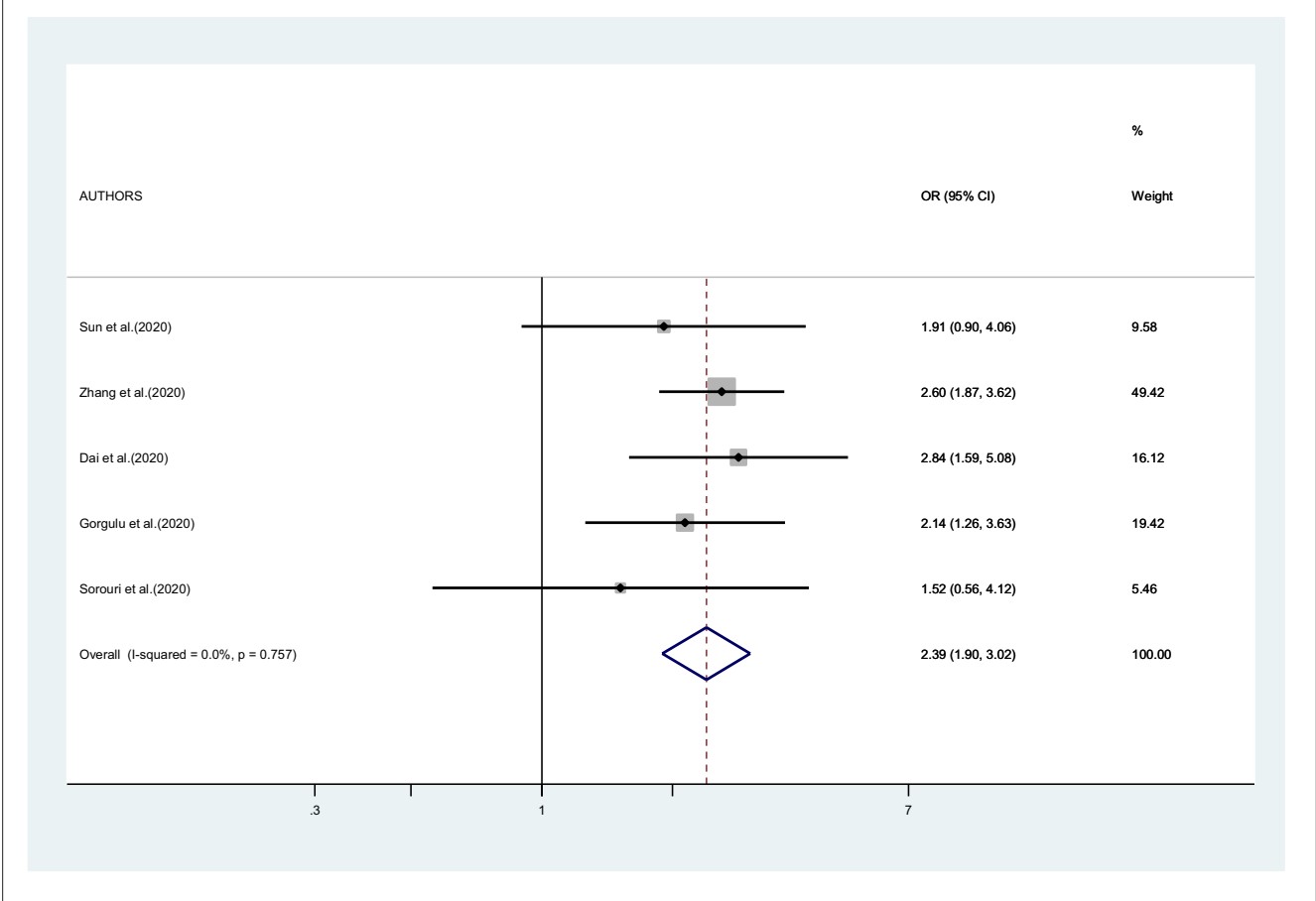

**Figure 3.** Forest plot – all types of cancer – outcome 2.

studies with results calculated on the basis of raw data. Furthermore, we considered the funnel plot and performed the Egger's regression asymmetry test to assess publication bias (*Egger et al., 1997*).

Finally, we conducted a cumulative meta-analysis, based on date of publication of subsequent studies.

Analyses were performed by STATA16 program (*StataCorp, 2019*), using specific commands metan, metabias, and metafunnel.

## Results

We identified a total of 3488 publications from the literature search and excluded 3 because they were duplicates. We screened the titles and abstracts of 3485 articles: we excluded 3145 of them because not relevant (*Figure 1*), and retained 340 articles as potentially eligible.

After reviewing the full-texts, we excluded 303 articles because these did not meet the inclusion criteria, and included the remaining 37 studies in the review: we finally included 35 of them in the quantitative synthesis.

Among the 35 studies, 30 reported results for all cancers combined, and 8 for hematologic neoplasms (3 of these reported both sets of results). Results for other specific cancers were sparse, and we could not conduct meta-analyses of them. Out of the 35 studies, 13 were from Europe, 11 from North America (all from the United States), and 11 from Asia (9 from China and 2 from Iran). Fifteen studies were considered good quality (CASP score>9.5), 19 studies were of moderate quality (9.5≥CASP score>6), whereas 1 was considered inadequate (CASP score≤6).

*Tables 1 and 2* show the details of the studies included in the analysis.

*Figures 2–4* report the results of the meta-analyses of studies of COVID-19 patients with all cancers combined compared to patients without cancer, for mortality, admission to ICU, and hospitalization or

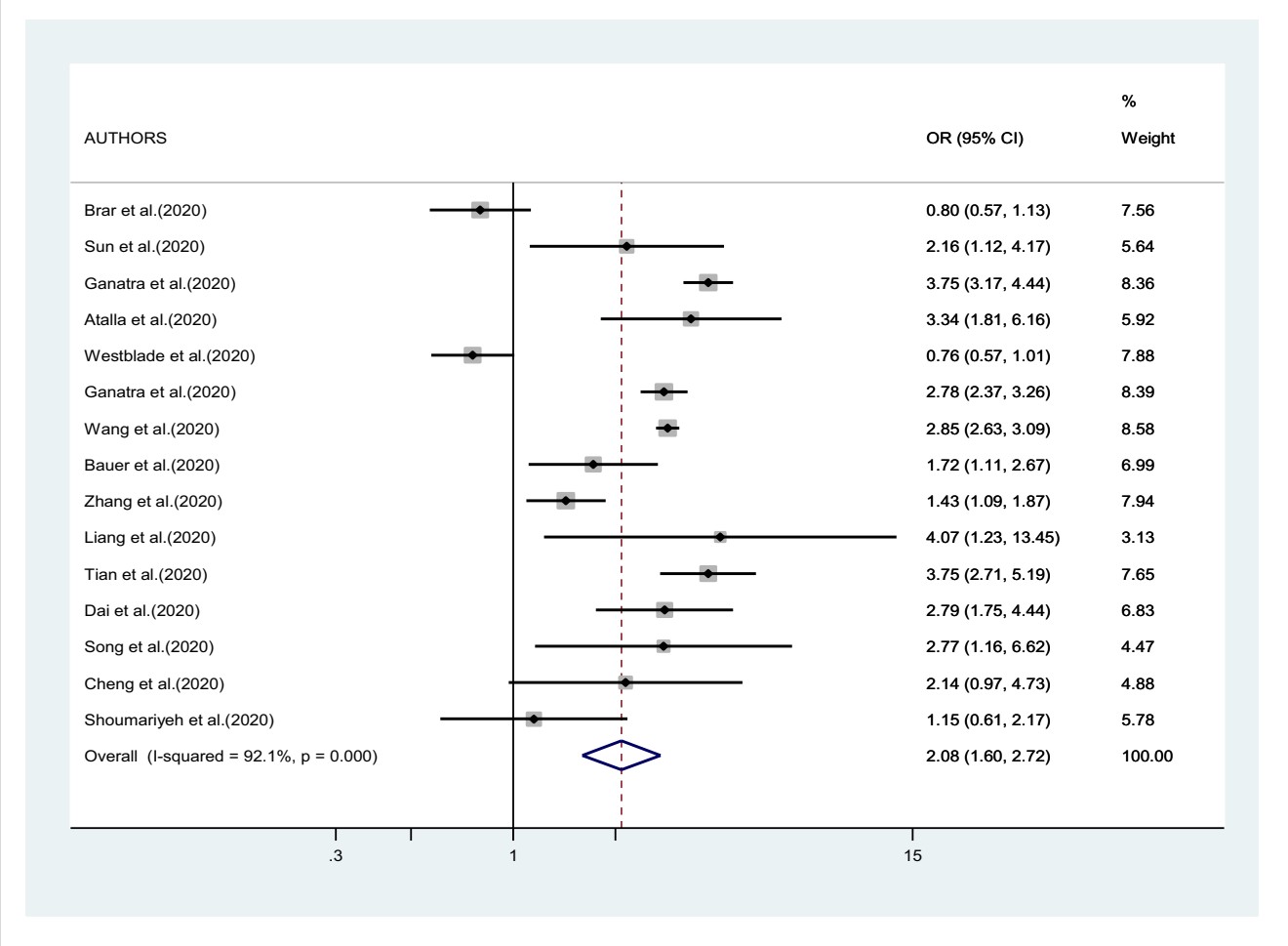

**Figure 4.** Forest plot – all types of cancer – outcome 3 or 4.

severity of COVID-19, respectively. The pooled OR for mortality was 2.32 (95% CI 1.82–2.94, $I^2$ 90.1%), that for ICU admission was 2.39 (95% CI 1.90–3.02, $I^2$ 0.0%), and that for hospitalization/severity of disease was 2.08 (95% CI 1.60–2.72, $I^2$ 92.1%).

In the analysis by geographic region (*Figure 2*), the association between SARS-CoV-2 infection and mortality in cancer patients was stronger, and less heterogeneous, in studies from Asia (OR 2.92; 95% CI 2.13–4.01, $I^2$ 37.8%) than in studies from either Europe (OR 2.37; 95% CI 1.65–3.40; $I^2$ 67.9%) or North America (OR 1.97; 95% CI 1.31–2.97; $I^2$ 95.9%, respectively). Too few studies were available on the other outcomes to justify a meta-analysis stratified by region of origin.

The cumulative meta-analysis, based on date of publication of subsequent studies of mortality (all types of cancer), showed a stronger association in the studies published before July 2020 than in studies published later (results not shown in detail).

As shown in *Figure 5*, we found no evidence of publication bias in the meta-analysis concerning mortality (p value of Egger's test 0.67). The number of studies included in the other meta-analyses was too low to yield meaningful results on publication bias.

In the sensitivity analysis based on QA, the pooled OR of mortality results of studies with acceptable quality was not different from that of results of good-quality studies: OR 2.25 (95% CI 1.73–2.94) versus OR 2.50 (95% CI 1.47–4.26). When we repeated the analysis after excluding one study at a time, we did not identify a major effect of any single study; in particular, the exclusion of the only study that suggested a negative association between cancer and mortality (*Harrison et al., 2020*) yielded a pooled OR of 2.41 (95% CI 1.95–2,99, $I^2$ 85.5%). The association with mortality was less pronounced in studies whose results were reported by the authors (OR 2.11; 95% CI 1.55–2.87) compared to studies

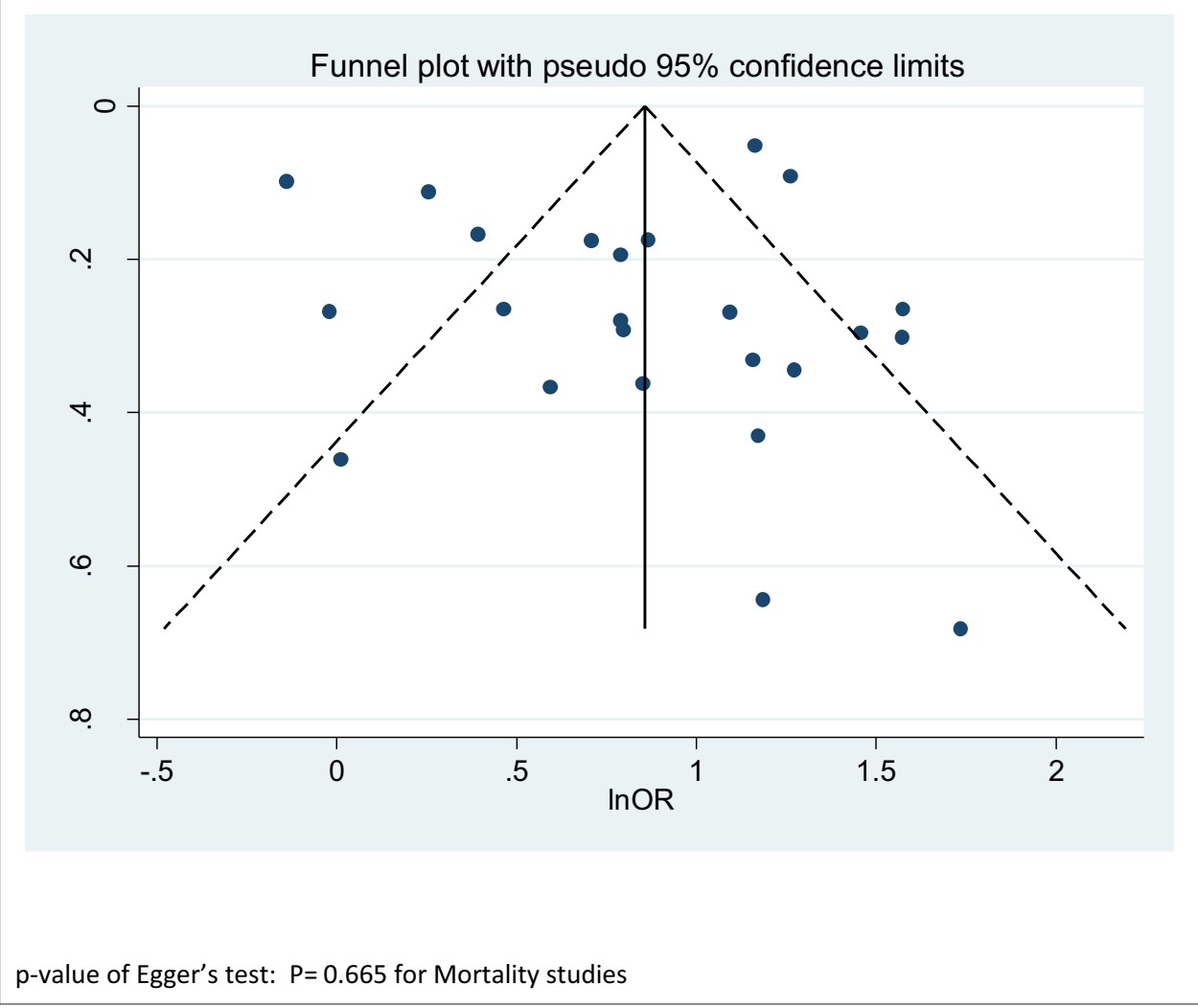

**Figure 5.** Funnel plot of Egger's test to assess publication bias – all types of cancer – outcome 1.

whose results were calculated by us (OR 2.66; 95% CI 1.97–3.60), although the difference was not statistically significant (*Figure 6*).

*Figure 7* presents the results of the meta-analysis of eight studies on mortality in patients with hematologic neoplasms. The pooled RR was 2.14 (95% CI 1.87–2.44, I² 20.8%). Results for other outcomes (admission to ICU, hospitalization, and severity of symptoms) were too sparse to conduct a meta-analysis.

## Discussion

Since the beginning of the SARS-CoV-2 pandemic, cancer patients affected by COVID-19 have been identified to be at increased risk of poor prognosis, together with other vulnerable categories of patients as those affected by cardiovascular disease, diabetes, kidney injury, obesity, or stroke (*Hu et al., 2020*).

However, how much SARS-CoV-2 infection resulted in more severe outcomes in cancer patients compared to patients without cancer and what caused their worse clinical course has not been fully clarified. In a previous editorial, some of us addressed the issue of the different interactions that COVID-19 and cancer may have (*Hainaut and Boffetta, 2021*). On one hand, it is interesting to study how COVID-19 evolves in patients with cancer, by assessing whether the infection in these patients has a more severe course than in a control group affected by the infection but without cancer. On the

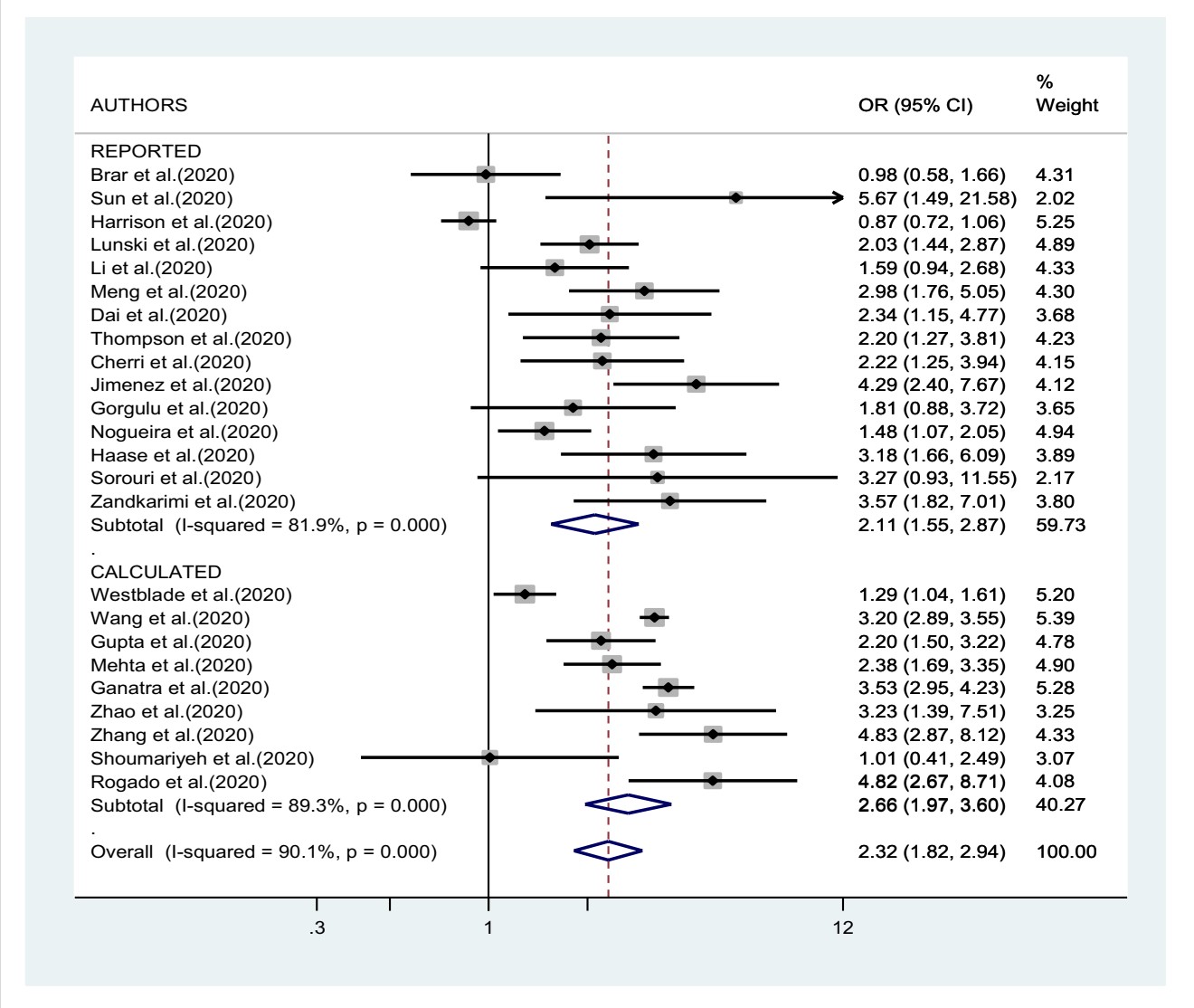

**Figure 6.** Forest plot – all types of cancer – outcome 1 – reported versus calculated OR.

other hand, it is important to identify the effects that the pandemic itself has determined in patients with cancer, including reduced access to treatment, delay in diagnosis for postponed screening, increased time between follow-up visits, and change in treatment organization. Acquiring more severe infection could be due to both components.

In this systematic review and meta-analysis, we focused on the effect that cancer had in patients with COVID-19 compared with those without cancer in terms of mortality, ICU access, and severity of COVID-19 (hospitalization or severity of symptoms). We found that patients with cancer and SARS-CoV-2 infection have a twofold higher risk of experiencing these adverse outcomes compared to non-cancer patients. Our results are in agreement with those of the meta-analysis by *Venkatesulu et al., 2020*, who included a smaller number of studies, mainly from China, and reported an OR of 2.54 (95% CI 1.47–4.42) for mortality in cancer patients with concurrent COVID-19, compared to non-cancer patients. Similar to our results, these authors also reported a stronger association in studies from China than in those from other regions. Our results are also similar to those by *Zhang et al., 2020a*, who reported a meta-analysis of five studies from China, yielding a meta-OR of 2.63, with limited heterogeneity. Compared to these early reports, we included more studies, which should lead to a more robust and precise risk estimate.

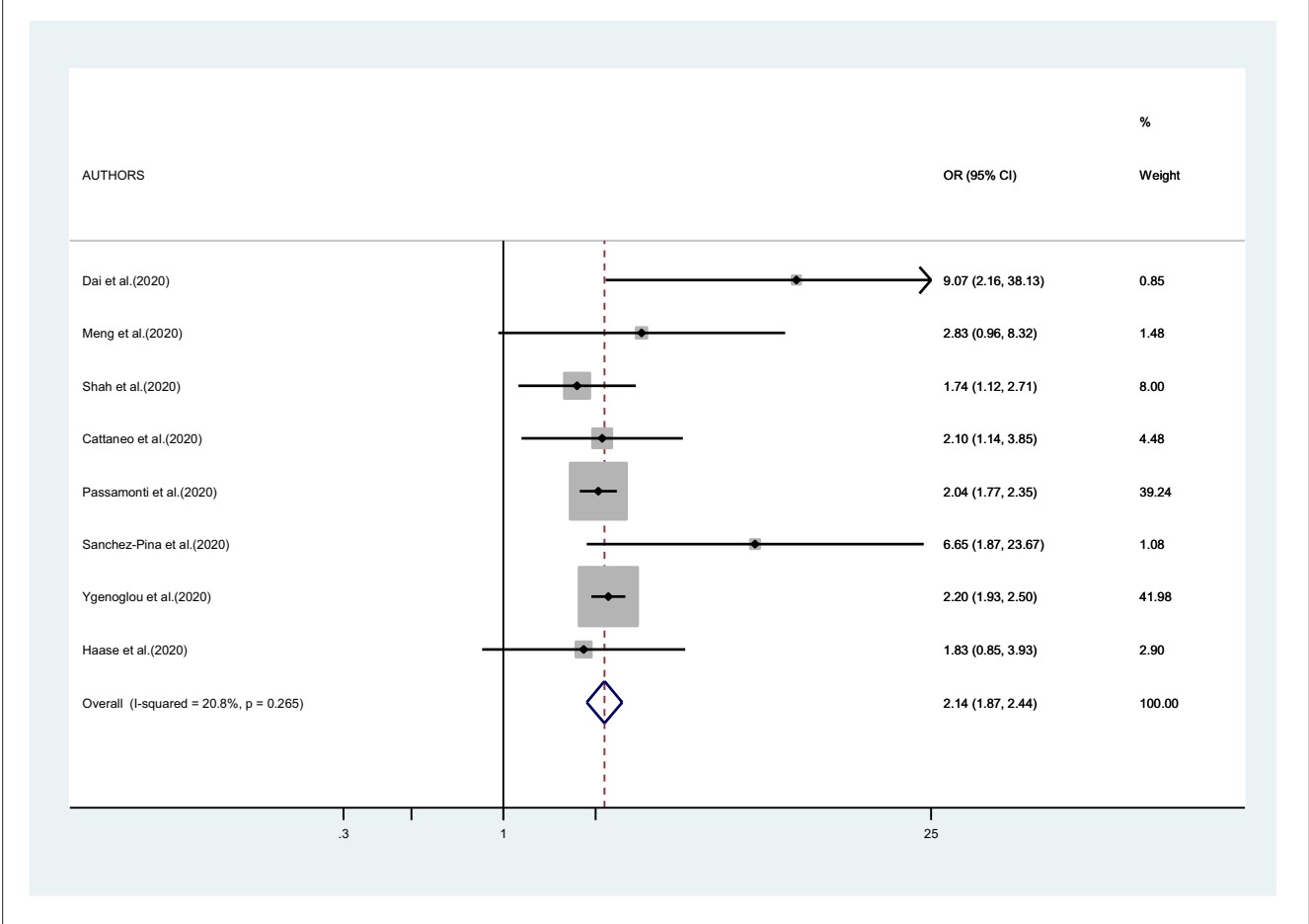

**Figure 7.** Forest plot – hematological neoplasms – outcome 1.

The online version of this article includes the following figure supplement(s) for figure 7:

**Figure supplement 1.** Forest plot – hematological neoplasms – outcome 2, 3, or 4.

The higher risk of mortality in studies from China compared to those from other countries could be explained by the fact that some of the studies from China were conducted during the very early phase of the infection, when diagnosis and treatment for SARS-CoV-2 might have been delayed, resulting in higher death rate. This interpretation is reinforced by the results of the cumulative meta-analysis that showed a stronger effect detected in the early studies compared to later studies.

Our summary results on the risk of ICU admission and severity of COVID-19 indicated a somehow weaker association than that reported by other authors. An early meta-analysis reported a threefold increase for ICU admission, an almost fourfold increase for a SARS-CoV-2 infection classified as severe, and a fivefold increase in being intubated (*ElGohary et al., 2020*). The fact that our values are lower might be explained by the inclusion of studies conducted when management of cancer patients with SARS-CoV-2 infection was more effective.

Immunosuppression and impaired T-cell response due to therapies may underlie the worse outcome in hematologic cancer diseases, even if some authors suggested that the attenuated inflammatory response in hematological patients can protect from severe COVID-19 morbidity (*Vijenthira et al., 2020*). The results of our meta-analysis confirm a higher mortality from COVID in patients with hematological neoplasms compared to non-neoplastic patients, with limited heterogeneity, with a pooled risk estimate similar to that for all cancers combined.

We were not able to derive pooled results for other specific cancers. Results for patients with hematologic and solid neoplasms were compared in some individual studies. In particular, *Desai*

*et al., 2021* reported a higher mortality in the former group, but the comparison was not adjusted for age and type of therapy.

Although our study provides the most precise measure to date of the effect of cancer in COVID-19 patients, it suffers from some limitations. Many studies included in our analysis did not provide results adjusted for important determinants such as sex, age, comorbidities, and therapy. As mentioned above, we were not able to analyze specific cancers other than hematologic neoplasms, because results were too sparse.

In conclusion, our meta-analysis confirms, by giving a more precise and accurate estimation, evidence to the hypothesis of an association in COVID-19 patients between cancer (and more specific hematologic neoplasm) and a worst outcome on mortality, ICU admission, and severity of COVID-19.

Future studies will be able to better analyze this association for different subtypes of cancer, and to evaluate whether the effects identified before vaccination are attenuated vaccinated patients.

## Additional information

### Funding
No external funding was received for this work.

### Author contributions
Giulia Di Felice, Giovanni Visci, Conceptualization, Data curation, Formal analysis, Investigation, Methodology, Project administration, Software, Supervision, Writing – original draft, Writing – review and editing; Federica Teglia, Marco Angelini, Data curation, Methodology, Visualization; Paolo Boffetta, Conceptualization, Methodology, Project administration, Supervision, Validation, Writing – review and editing

### Author ORCIDs
Giovanni Visci http://orcid.org/0000-0003-2246-2321
Federica Teglia http://orcid.org/0000-0003-3188-1632

### Ethics
Human subjects: (a) All methods were carried out in accordance with relevant guidelines and regulations. (b) The study was considered exempt and the informed consent was not deemed necessary given the nature of the study.

### Decision letter and Author response
Decision letter https://doi.org/10.7554/eLife.74634.sa1
Author response https://doi.org/10.7554/eLife.74634.sa2

## Additional files

### Supplementary files
- Supplementary file 1. Protocol.
- Supplementary file 2. Quality Assessment.
- Reporting standard 1. PRISMA checklist.

### Data availability
All data generated or analysed during this study are included in the manuscript and supporting file. Dataset has been deposited on Dryad (https://doi.org/10.5061/dryad.00000004q).

The following dataset was generated:

| Author(s) | Year | Dataset title | Dataset URL | Database and Identifier |
|---|---|---|---|---|
| Visci G, Di Felice G, Teglia F, Angelini M, Boffetta P | 2021 | Data from: Effect of SARS-CoV-2 infection on outcome of cancer patients: A systematic review and meta-analysis of studies of unvaccinated patients | http://dx.doi.org/10.5061/dryad.00000004q | Dryad Digital Repository, 10.5061/dryad.00000004q |

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
