## [Editor Report]

The authors conducted a systematic review and baseline meta-analysis of studies on the impact of SARS-Cov-2 infection on morbidity and mortality among cancer patients not previously vaccinated against the virus. This analysis serves as benchmark for forthcoming work on the same outcomes among vaccinated cancer patients, which as a whole will assist the development of cancer care guidelines.

---

## [Decision Letter]

**Decision letter after peer review:**

Thank you for submitting your article "Effect of SARS-CoV-2 infection on outcome of cancer patients: A systematic review and meta-analysis of studies of unvaccinated patients" for consideration by *eLife*. Your article has been reviewed by 3 peer reviewers, one of whom is a member of our board of Reviewing Editors and the evaluation has been overseen by a Senior Editor. The following individual involved in review of your submission has agreed to reveal their identity: Sina Azadnajafabad (Reviewer #1).

Essential revisions:

Thank you for submitting your manuscript to the Special issue of *eLife*. Based on our editorial evaluation and the comments of our peer reviewers, I regret to inform you that we will not accept your manuscript for publication in the current form. However, we would reconsider your manuscript following revision and modification addressing the reviewer's concerns.

The reviews converge at some points that must be addressed in detail in your revision. The first point has to do with the statistical methods and whether the results can be interpreted correctly. Specifically, on the inclusion of non-cancer controls with uninfected cancer patients in the reference group; the estimation of the pooled odds ratios for mortality, ICU admissions and hospitalization, and severe COVID-19 disease between infected cancer patients versus uninfected cancer patients and non-cancer controls, respectively, should be separate and clearly delineated. Second, additional clarity is needed describing how the study quality assessment tool was adapted to your particular research question and source study designs. Third, restriction of the systematic review to studies published in 2020 limits the novelty and timeliness of this meta-analysis; the reviewers recommend expanding the scope to include more recent studies with comparisons between vaccinated and unvaccinated cancer patients.

When revising your manuscript, please consider all issues mentioned in the reviewers' comments carefully: please outline every change made in response to their comments and provide suitable rebuttals for any comments not addressed. Please note that your revised submission may need to be re-reviewed.

*Reviewer #1:*

I read the manuscript with interest as the topic was exciting and the investigated idea falls into my interested fields. However, the idea and conducted review and synthesized results are by some means out-of-date and many similar published papers on this topic are available now; however, with some smaller size in review. I recommend expanding the review to a more updated population like the vaccinated ones and also extending the period of search to find more recently published literature, to effectively add to the body of evidence in the vast of cancers.

*Reviewer #2:*

Di Felici et al., investigated the outcomes of people with cancer who are infected by SARS-CoV-2 or develop COVID-19, carrying out a systematic review of studies published to 30 December 2020. They suggest that people with cancer have a pooled odds ratio of 2.32 for mortality, 2.39 for ICU admission, and 2.08 for disease severity and hospitalisation. In the abstract, this is concluded to mean that there is a "two-fold increased risk of adverse outcomes (mortality, ICU admission and severity of COVID-19) in unvaccinated cancer patients infected with SARS-CoV-2 compared to uninfected patients". Unfortunately, the methods used in this review in its current form have severe methodological shortcomings that render the results uninterpretable.

Key issue 1: pooling fundamentally different comparisons

In the Discussion, the authors state that they analysed "the effect that SARS-CoV-2 infection had in patients with cancer compared with those without cancer in terms of mortality, ICU access, and severity of COVID-19 (hospitalization or severity of symptoms)", in a direct conflict with the comparison described in the Abstract.

Looking at the studies included in the same meta-analysis, the authors evidently combine comparisons of

(A) people with COVID-19 and cancer to people with COVID-19 but with no cancer, and

(B) people with COVID-19 and cancer to people with no COVID-19 but with cancer.

This is incorrect as (A) and (B) are fundamentally different: the key difference in (A) relates to cancer status, while the key difference in (B) relates to COVID-19 status.

As an analogy, this would be similar to pooling comparisons of people

(A1) age<40 and male to age<40 and female, and

(B1) age<40 and male to age>40 and male.

In summary, this means the meta-analyses combine very different types of risk comparisons and the results of the review are not interpretable.

Key issue 2: quality assessment limitations

In the second key limitation, the authors state results of a quality assessment, but do not describe how the quality assessment tool was adapted to this particular research question (e.g. what was deemed "acceptable" recruitment of the cohort, "accurate" approaches to exposure and outcome measurement, "all important confounding factors", "complete enough follow-up" etc). Notably, the tool they used is specific to cohort studies, while some of the included studies have a case-control design (e.g. Shoumariyeh et al.,), so this tool is not applicable and quality issues related to case-control studies would not be assessed correctly.

As thorough risk of bias assessment is a key feature required for systematic reviews, this is an important shortcoming further reducing the quality of this study.

The two key issues described above would need to be addressed for the study to be valid.

*Reviewer #3:*

This study presents results from a systematic review and meta-analysis of studies published in 2020 showing increased risk of adverse clinical outcomes, including mortality, ICU admissions, severe COVID-19 disease, and hospitalizations, among unvaccinated cancer patients infected with COVID-19. While this represents an important question for populations and regions where vaccination rates remain low and cases with COVID-19 disease are prevalent, the focus on the period prior to the widespread introduction of vaccines reduces the study's relevance. In addition, the lack of clarity on the methods and the conflation of uninfected cancer patients with non-cancer controls makes it difficult to draw clear conclusions.

The question of the impact of COVID-19 infection on clinical outcomes in unvaccinated cancer patients is an important question for populations and regions where vaccination rates remain low and cases with COVID-19 prevalent. The relevance of study would be strengthened, however, by including more recent studies that compared vaccinated to unvaccinated cancer patients. In addition, the comparisons between infected and uninfected cancer patients and non-cancer controls should be clearly delineated.

---

## [Author Response]

Essential revisions:The reviews converge at some points that must be addressed in detail in your revision. The first point has to do with the statistical methods and whether the results can be interpreted correctly. Specifically, on the inclusion of non-cancer controls with uninfected cancer patients in the reference group; the estimation of the pooled odds ratios for mortality, ICU admissions and hospitalization, and severe COVID-19 disease between infected cancer patients versus uninfected cancer patients and non-cancer controls, respectively, should be separate and clearly delineated.

We actually found that there was some confusion about the two comparison groups in our meta-analysis (ie COVID patients with cancer VS COVID patients without cancer); we therefore made the changes in the text and also in the title of our Paper. We uploaded a new version with the changes highlighted and also a clean version.

Second, additional clarity is needed describing how the study quality assessment tool was adapted to your particular research question and source study designs.

We uploaded the re-adapted Quality Assessment for the different study designs (the case-control studies had only one less question at the start); it is detectable as Supplementary file 2.

Third, restriction of the systematic review to studies published in 2020 limits the novelty and timeliness of this meta-analysis; the reviewers recommend expanding the scope to include more recent studies with comparisons between vaccinated and unvaccinated cancer patients.

While we agree with the reviewers, we think that including a systematic review and meta-analysis of studies conducted in vaccinated cancer patients would fundamentally transform our manuscript. The number of studies to review would increase dramatically, with consequences in terms of tables and text, and we would require a few months to complete this task. In fact, we have planned to do this analysis and include it in a second manuscript, and we decided to wait until early spring 2022 to identify papers covering the experience of the first year after vaccination. We think our results restricted to the pre-vaccination experience are valuable per se, as they captures a unique experience in medicine, that hopefully will not be replicated, at least for SARS-CoV-2.